# Integrated Lipidomics and Transcriptomics Analyses Reveal Key Regulators of Fat Deposition in Different Adipose Tissues of Geese (*Anser cygnoides*)

**DOI:** 10.3390/ani14131990

**Published:** 2024-07-05

**Authors:** Maodou Xu, Yaoyao Zhang, Yang Zhang, Qi Xu, Yu Zhang, Guohong Chen

**Affiliations:** 1College of Animal Science and Technology, Yangzhou University, Yangzhou 225009, China; xumaodou0106@163.com (M.X.); yaoyaochn@outlook.com (Y.Z.); zyang@yzu.edu.cn (Y.Z.); xuqi@yzu.edu.cn (Q.X.); 2Key Laboratory for Evaluation and Utilization of Livestock and Poultry Resources (Poultry), Ministry of Agriculture and Rural Affairs, Beijing 100176, China

**Keywords:** geese, gene regulation, fat deposition, lipidomics, transcriptomics

## Abstract

**Simple Summary:**

Intramuscular fat (IMF) content is a crucial factor that affects the quality of goose meat. The deposition of fat in different adipose tissues of geese is interconnected. Specifically, an increase in IMF deposition may also lead to increased deposition of abdominal and subcutaneous fat, which reduces feed efficiency and affects reproductive performance. This study examined the heterogeneity in lipid composition among the three adipose tissues, IMF, abdominal fat, and subcutaneous fat, and explored the key regulatory factors affecting them. The aim was to provide valuable insights into the selection and improvement of goose meat through targeted fat deposition.

**Abstract:**

The fat deposition of different adipose tissues is widely recognized as correlated, with distinct effects on meat quality traits and reproductive performance in poultry. In this study, we utilized lipidomics and transcriptomics analyses to investigate the heterogeneity and regulators of intramuscular fat (IMF), abdominal fat (AF), and subcutaneous fat (SF) in geese. Lipidomic profiling revealed 165, 129, and 77 differential lipid molecules (DLMs) between AF vs. IMF, SF vs. IMF, and SF vs. AF, respectively, with 47 common DLMs identified between AF vs. IMF and SF vs. IMF. Transcriptomic analysis identified 3369, 5758, and 131 differentially expressed genes (DEGs) between AF vs. IMF, SF vs. IMF, and SF vs. AF, respectively, with 2510 common DEGs identified between AF vs. IMF and SF vs. IMF. The KEGG results indicate that DLMs were predominantly enriched in glycerophospholipid and glycerolipid metabolism pathways, while DEGs were primarily enriched in metabolic pathways. Pearson correlation analysis identified *FABP4*, *LPL*, *PLCB1*, *DSE*, and *PDE5A* as potential factors influencing fat deposition. This study elucidates the heterogeneity and regulatory factors of different adipose tissues in geese, offering new insights for targeted improvements in goose meat quality and production efficiency.

## 1. Introduction

According to the United Nations Food and Agriculture Organization, China dominates global goose meat production, contributing to 95.8% of the world’s total, establishing itself as the leading producer and consumer of goose meat worldwide [1]. Geese are characterized by a longer growth cycle and superior nutrient content compared to chickens and ducks [2]. The rich protein content in goose meat makes it a crucial source of essential amino acids and beneficial unsaturated fatty acids, offering significant benefits to human health and well-being [3]. However, geese, as waterfowl, possess a robust capacity for fat deposition, wherein a significant amount of fat accumulates, particularly in the abdominal region, serving as an insulating layer to prevent heat loss. The percentage of abdominal fat in 16-week-old female White Kołuda geese was 5.16%, while the percentage of skin with subcutaneous fat was 21.03% [4]. Excessive abdominal fat (AF) and subcutaneous fat (SF) deposition not only reduces feed conversion efficiency and carcass yield but also affects reproductive functions. Due to the correlation between fat deposition in different adipose tissues, a decrease in AF and SF leads to a decrease in intramuscular fat (IMF) [5]. IMF is identified as a primary determinant of meat quality [6]. The oxidation of IMF exhibits a strong correlation with meat color determined by myoglobin, which reflects the meat freshness and significantly influences consumer preferences [7]. IMF also influences muscle fiber properties, thereby enhancing muscle tenderness [8]. Moreover, fat and protein contribute significantly to volatile compound generation during cooking [9]. Therefore, identifying regulators involved in fat deposition in different adipose tissue of geese serves as a target for directional fat deposition, which can reduce abdominal and subcutaneous fat deposition without compromising meat quality.

Adipose tissue is formed by the proliferation and enlargement of adipocytes, which are divided into white adipocytes and brown adipocytes. White adipocytes primarily function in energy storage, mobilization of fatty acids to release energy, and buffering the impact on the body, whereas brown adipocytes dissipate energy through nonshivering thermogenesis [10,11,12]. Avian adipose tissue primarily consists of white adipose tissue [13]. In poultry such as chickens, ducks, and geese, fat predominantly deposits in muscles, subcutaneous tissue, and the abdominal region. In addition to correlations, variability exists in the deposition of different adipose tissues [14]. AF deposition primarily involves fatty acid synthesis within mitochondria, whereas IMF synthesis relies more on gluconeogenesis and energy metabolism [5]. Notably, SF deposition occurs at a faster rate compared to IMF deposition, resulting in higher subcutaneous fat content [15]. Adipocytes of SF can be observed earlier compared to those of AF during embryonic stages. However, adipocytes of AF exhibit a stronger capacity for fat storage, continuously increasing during the growth period compared to SF [16]. Various candidate genes influence fat deposition, and their expression and abundance vary across different parts. In WenChang chickens, IMF deposition correlates with genes involved in acetate and citrate metabolism, including *GAPDH*, *LDHA*, *GPX1*, and *GBE1*, while AF deposition correlates with genes involved in acetyl-CoA and glycerol metabolism, such as *FABP1*, *ELOVL6*, *SCD*, and *ADIPOQ* [17]. Similarly, in geese, regulators accelerating fat deposition have been identified, with *ALDOA* and *GPX1* exhibiting higher expression levels in the breast muscle of Xingguo Gray geese. Additionally, a positive correlation was observed between ALDOA, GPX1, fat content, and triglyceride (TG) content, with *ALDOA* and *GPX1* promoting IMF accumulation by positively regulating the synthesis of acetate and lactate [18]. Fat deposition in goose liver is increased by adding betaine, which upregulates LPL mRNA expression to regulate TG storage in the liver rather than adipocytes [19]. In Yangzhou geese, key genes involved in adipogenesis, including *PPARγ*, *C/EBPα*, *FABP4*, and *LPL* are predominantly expressed in subcutaneous adipocytes, whereas C/EBPβ is highly expressed in IMF cells [16]. Overall, these findings indicate the existence of different regulators in different adipose tissues, but how these regulators contribute to tissue heterogeneity in fat deposition by affecting lipid metabolism remains to be studied.

Lipidomics, a rapidly evolving analytical tool based on liquid chromatography-mass spectrometry (LC-MS), has gained prominence in recent years for the comprehensive study of lipid composition. It has demonstrated considerable success in the realm of food and nutrition science. Specifically, lipidomics has proven effective in pinpointing specific lipids within different muscle regions of pork and beef, as well as examining lipid dynamics during fermentation in pecan nuts and mandarin fish, among other applications [20,21,22]. Notably, in the medical field, lipidomics has emerged as a valuable avenue for research initiation, leveraging its ability to compare lipid composition disparities between healthy and diseased individuals. This application holds promise for identifying potential lipid markers [23,24]. Within the realm of livestock and poultry research, the primary focus of lipidomics has centered on discerning variations in fat deposition across different adipose tissues [25,26]. This study represented a pioneering effort in applying ultra-high-performance liquid chromatography-electrospray ionization–tandem mass spectrometry (UPLC-ESI-MS/MS)-based lipidomics to comprehensively profile the overall lipid compositions of IMF, AF, and SF in Zhedong white geese. Key differential lipids were characterized, and the potential metabolic pathways underpinning lipid heterogeneity in different adipose tissue were explored. Furthermore, transcriptomics analysis was employed to scrutinize differentially expressed genes (DEGs) and their functional enrichment in different adipose tissue, providing insights into the genetic regulation of fat deposition. By integrating transcriptomic and metabolomic analyses, we systematically explored candidate functional genes and signaling pathways regulating the deposition of IMF, AF, and SF in geese, aiming to comprehensively depict the regulatory network of fat differentiation in different adipose tissue. Our primary objective was to elucidate the molecular regulatory mechanisms, heterogeneity, and specific deposition control targets of fat in different adipose tissue, thus providing a scientific basis for efficient and high-quality production of goose meat.

## 2. Materials and Methods

### 2.1. Animal Handling

All animal experimental protocols used in the present study were approved by the Yangzhou Institutional Animal Committee (Approval Number: 132-2022; 29 May 2022). The procedures were performed in accordance with the Regulations for the Administration of Affairs Concerning Experimental Animals (Yangzhou University, Yangzhou, China, 2012) and the Standards for the Administration of Experimental Practices (Jiangsu, China, 2008).

A total of six healthy 70-day-old Zhedong white geese (three males and three females) from Changzhou Siji Poultry Industry Co., Ltd., Suqian, China, were used for the construction of lipidome profiles and transcriptome sequencing. All geese shared identical genetic backgrounds and dietary conditions and were raised in the same environmental conditions. All geese were fed a standardized commercial diet, and the ingredients and chemical composition of the diet are detailed in Appendix A [27], adhering to the national standards of the People’s Republic of China for Zhedong white goose (GB/T 36178-2018) [28]. This feeding regimen was maintained until they reached 70 days of age.

### 2.2. Sample Collection

Sample collection was conducted in the laboratory for the six randomly selected geese. Following a 12-h fasting period, they were humanely euthanized by exsanguination, and the pectoralis major muscles, subcutaneous fat tissue, and abdominal fat tissue were collected. Subsequently, the AF and SF were weighed, and the relative weight of fat tissues was measured. The collected samples were then rapidly frozen in liquid nitrogen. The samples were divided into two parts: one part was stored at −80 °C for RNA extraction, lipidomic and transcriptomic analyses, and the other part was fixed with 4% paraformaldehyde for hematoxylin-eosin (HE) staining.

### 2.3. Lipid Extraction

The AF (*n* = 6), SF (*n* = 6), and pectoralis samples (*n* = 6) were utilized for lipid extraction and lipidomic analysis. The lipid extraction procedure was conducted as follows: (1) Samples were thawed on ice, and 20 mg of the thawed sample was transferred to a corresponding 2 mL centrifuge tube. (2) To the 2 mL centrifuge tube, 1 mL of lipid extraction solution (including methanol, methyl tert-butyl ether (MTBE), and internal standard mixture) and 2 steel balls were added. (3) The samples were homogenized by crushing in a ball mill (MM400; Retsch, Haan, Germany). (4) After removing the steel ball, the homogenate was subjected to vortex oscillation for 2 min in a multitube vortex oscillator (MIX-200, Shanghai, China) and ultrasonication for 5 min in an ultrasonic cleaner (KQ5200E, Kunshan, China) with the addition of 200 μL ultra-pure water to the centrifuge tube. (5) The samples were mixed by vortexing for 1 min, then centrifuged at 12,000 rpm for 10 min at 4 °C (5424R, Eppendorf, Hamburg, Germany). (6) A new 1.5 mL centrifuge tube was prepared and labeled, and 200 μL of the clear supernatant solution from the previous step was aspirated into the corresponding 1.5 mL centrifuge tube, followed by concentration in a centrifugal concentrator (CentriVap, Kansas, MO, USA). (7) To the resulting concentrated powder, 200 μL of lipid reconstitution solution was added, and the mixture was subjected to UPLC-ESI-MS/MS analysis.

### 2.4. Lipidomic Assay

Lipid contents were detected using the AB Sciex QTRAP 6500 LC-MS/MS platform and analyzed by MetWare (http://www.metware.cn/, accessed on 15 November 2022). The instrumental system for data acquisition included ultra performance liquid chromatography (UPLC) (ExionLC™ AD, https://sciex.com.cn/, accessed on 15 November 2022) and tandem mass spectrometry (MS/MS) (QTRAP^®^ 6500+, https://sciex.com.cn/, accessed on 15 November 2022). The analytical conditions were as follows.

UPLC: Lipid separation was performed on a 2.6 μM, 2.1 mm × 100 mm i.d. column (Thermo Accucore™C30 column). The column temperature was maintained at 45 °C, and the flow rate was set at 0.35 mL/min. A 2 μL sample was injected from the autosampler, subjected to gradient elution in mobile phase A (acetonitrile/water; 60/40, V/V) and mobile phase B (acetonitrile/isopropanol; 10/90, V/V) containing 0.1% formic acid and 10 mmol/L formic acid ammonium. The elution gradient was set as follows: 20% B increased to 30% B at 0–2 min; increased to 60% B at 2–4 min; increased to 85% B at 4–9 min; equilibration with 90% B at 9–14 min; increased to 95% B at 14–15.5 min; equilibration with 95% B at 15.5–17.3 min; decreased to 20% B at 17.3–17.5 min; and finally, equilibration with 20% B at 17.5–20 min.

ESI-MS/MS: The effluent was directed alternately to an ESI-triple quadrupole-linear ion trap (QTRAP)-MS. The QTRAP^®^ 6500+ LC-MS/MS System, equipped with an ESI Turbo Ion-Spray interface, performed LIT and triple quadrupole (QQQ) scans under the following conditions: ESI temperature 500 °C, ion spray voltage 5.5 kv in positive mode, −4.5 kv in negative mode, ion source gas1, 45 psi, gas2, 55 psi, and curtain gas, 35 psi. In the triple quadrupole, each ion pair was scanned and detected based on optimized declustering potential (DP) and collision energy (CE).

Qualitative analysis of lipids within the 18 samples was conducted using the lipid database information established by MetWare Biotechnology Co., Ltd., Wuhan, China. Subsequently, MultiQuant^TM^ software 3.0.2 was employed to calibrate the chromatographic peaks detected for each substance in different samples, ensuring quantitative accuracy. The peak area in the chromatogram represented the relative content of each substance in its respective sample.

### 2.5. Lipidomics Data Analysis

Lipid content data were processed using unit variance scaling (UV). The orthogonal partial least squares-discriminant analysis (OPLS-DA) was conducted using the OPLSR. Anal function in the MetaboAnalystR package of R software 4.3.2. OPLS-DA integrates orthogonal signal correction (OSC) and partial least squares-discriminant analysis (PLS-DA) to identify differential lipids by eliminating unrelated variances. Variable importance in projection (VIP) values were extracted based on the OPLS-DA results. Differential lipids were selected based on variable importance in projection (VIP > 1) and the *p*-value from univariate analysis (*p* < 0.05) [26].

### 2.6. RNA Extraction, Library Preparation, and Sequencing

Total RNA was extracted from each sample. The integrity and potential contamination of RNA were assessed through agarose gel electrophoresis, and the purity was verified using the NanoPhotometer spectrophotometer (IMPLEN, Westlake Village, CA, USA). Qualified samples were subjected to library construction. Total RNA with a quantity ≥1 μg was utilized for cDNA library preparation using Illumina’s NEBNext UltraTM RNA Library Prep Kit (NEB, Ipswich, MA, USA). PolyA-tailed mRNA was enriched using Oligo (dT) magnetic beads. The obtained mRNA was randomly fragmented with divalent cations in the NEB Fragmentation Buffer. Using the fragmented mRNA as a template and random oligonucleotides as primers, the first cDNA strand was synthesized in the presence of M-MuLV reverse transcriptase. The RNA chain was degraded by RNaseH, and the second cDNA strand was synthesized using dNTPs as substrates in the presence of DNA polymerase I. The purified double-stranded cDNA underwent end repair, A-tailing, and adapter ligation. Fragments around 200 bp were selected using AMPure XP beads (Beckman Coulter, Beverly, CA, USA), followed by PCR amplification, and purification of PCR products, resulting in the construction of the library.

### 2.7. Transcriptome Data Analysis

The raw data is typically provided in fastQ format, encompassing sequence information of the sequencing fragments along with corresponding sequencing quality details. Employing fastp for raw data filtration involves the removal of paired reads where the N content exceeds 10% of the bases in that read and the elimination of paired reads containing low-quality bases (Q ≤ 20) exceeding 50% of the bases in that read. This process aims to obtain high-quality clean reads.

The cDNA libraries were sequenced on the Illumina sequencing platform by Metware Biotechnology Co., Ltd. (Wuhan, China). DESeq2 was utilized for inter-group differential expression analysis of samples, with the criteria for selecting DEGs set at |log2Fold Change| ≥ 1 and *p* < 0.05 [25]. Subsequently, featureCounts was used to calculate the gene alignment, and then calculate the fragments per kilobase of transcript per million fragments mapped (FPKM) of each gene based on the gene length. The FPKM values represent the relative abundance of genes in the sample, expressed in units of fragments per million base pairs.

### 2.8. Gene Expression Verification

To validate the accuracy of the transcriptomic results, 7 (AF vs. IMF), 6 (SF vs. IMF), and 4 (AF vs. SF) genes were randomly selected from each group for RT-qPCR verification. Total RNA from the fat tissues was extracted using the TRIZOL method (Tiangen Biotech Co., Beijing, China), following the manufacturer’s instructions. The FastKingRT kit (Tiangen Biotech Co., Beijing, China) was employed for the synthesis of the first-strand cDNA. Specific primers for quantitative real-time PCR (RT-qPCR), designed using Primer 5, are listed in Appendix A. The SuperReal PreMix Plus (SYBR Green) kit (Tiangen Biotech Co., Beijing, China) was utilized with ACTIN as the reference gene, and the RT-qPCR was performed on the CFX 96 Real-Time System (Bio-Rad, Hercules, CA, USA). Gene expression levels were calculated using the 2^−∆∆Ct^ method.

### 2.9. Statistical Analysis

GraphPad Prism 9 software was employed for the differential analysis of all cytomorphologic data. SPSS 26.0 software was used for data analysis between two sample groups. After checking for homogeneity of variances, a one-way analysis of variance was conducted for significance testing. *p* < 0.05 was considered statistically significant. All data are presented as mean ± standard deviation (SD). Pearson correlation analysis of differential lipids and genes was conducted using the Corrplo package, and a correlation network graph was plotted. Correlation coefficients results examined were interpreted as “very weak” (|r|≤0.19), “weak” (0.20≤|r|≤0.39), “moderate” (0.40≤|r|≤0.59), “strong” (0.60≤|r|≤0.79), or “extremely strong” (0.80≤|r|≤1.00) [29]. The Kyoto Encyclopedia of Genes and Genomes (KEGG) Compound Database (http://www.kegg.jp/kegg/compound/, accessed on 15 November 2022) was employed to identify metabolic pathways associated with differential lipid molecules and genes. Pathways with *p* < 0.05 were considered significant.

## 3. Results

### 3.1. Morphological Differences among Different Adipose Tissues

HE staining was applied to IMF, AF, and SF of 70-day-old geese. Abundant unilocular adipocytes were observed in both the abdominal and subcutaneous tissues, while only a small number of unilocular adipocytes were present in the muscles. Furthermore, lipid and TG content of AF and SF was significantly higher than IMF. The diameter of adipocytes in the AF and SF was significantly larger than IMF, whereas the number of adipocytes in the AF and SF was significantly lower than IMF (Figure 1).

### 3.2. Lipid Profile of Different Adipose Tissues

Lipid qualitative and quantitative analyses were conducted on the IMF, AF and SF of geese. The lipidic constituents were classified into eight major classes, namely, fatty acyls (FAs), glycerolipids (GLs), glycerophospholipids (GPs), sphingolipids (SPs), sterol lipids (STs), prenol lipids (PLs), saccharolipids (SLs) and polyketides (PKs). In this study, a total of 36 lipid subclasses were identified from 6 categories, and the overall quantity of lipid molecules among the three adipose tissues was similar. The IMF contained 839 lipid species, the AF contained 827 lipid species, and the SF contained 837 lipid species (Figure 2A). The lipid subclasses are predominantly composed of triglycerides (TG), phosphatidylethanolamine (PE), diacylglycerol (DG), phosphatidylcholine (PC), phosphatidylserine (PS), and phosphatidylglycerol (PG), constituting 73% of the total lipid molecules. Among these lipid categories, GLs contained the highest number of molecular species, and GPs encompassed the greatest number of subclasses (Figure 2B). Although quantity of lipid molecules was similar in the three adipose tissues, there were differences in content composition at the subclass level. Comparative analysis at the subclass level among the three fat deposits revealed that IMF has higher levels of coenzyme Q (CoQ) compared to AF and SF. Free fatty acid (FFA) content was higher in AF and SF compared to IMF. AF showed higher levels of diacylglycerol (DG) and lysophosphatidic acid (LPA) than IMF and SF. SF exhibited higher levels of lysophosphatidylinositol (LPI) than IMF and AF (Appendix A).

### 3.3. Differential Lipid Molecules of Different Adipose Tissues

OPLS-DA was employed to ascertain the correlation between different molecular components and identify differentiating lipids. The separation patterns of individual samples for each of the three components are depicted in Figure 3A. For the comparison between IMF and AF: R^2^X = 0.585, R^2^Y = 1.000, and Q^2^ = 0.653. For the comparison between IMF and SF: R^2^X = 0.49, R^2^Y = 0.995, and Q^2^ = 0.586. For the comparison between AF and SF: R^2^X = 0.617, R^2^Y = 0.998, and Q^2^ = 0.583. All three Q^2^ values exceeded 0.5, indicating effective models and confirming the existence of differences in different adipose tissues, consistent with previous reports [30]. Using VIP > 1 and *p* < 0.05 as selection criteria, the lipid molecules were analyzed in three pairwise combinations, and differential lipid molecules (DLMs) were identified. As shown in Figure 3B comparing AF with IMF, 165 DLMs were identified, including 88 downregulated and 77 upregulated in AF compared with that in IMF. The five most significantly distinct lipid molecules included PE (O-16:1_24:4), PC (18:0_20:4), PE (O-18:1_22:2), PC (16:0_20:4), and Cer (16:1/26:0). When comparing SF with IMF, 129 DLMs were identified, with 45 downregulated and 84 upregulated in SF compared with that in IMF. The five most significantly distinct lipid molecules included TG (15:1_15:1_19:2), TG (10:0_16:0_16:1), PS (21:2_20:4), Cer (d16:0/21:0), and PC (O-16:0_20:4). When comparing SF with AF, there were 77 significantly DLMs, including 28 downregulated and 49 upregulated in SF compared with that in AF. The upregulated lipids in SF were mainly GPs. The five most significantly distinct lipid molecules based on VIP selection were Cer (d21:2/38:1(2OH)), PE (O-16:1_24:4), LNAPE (18:2/N-18:2), TG (17:0_18:0_18:1), and PI (18:0_22:4). The lipid molecules exhibiting the greatest disparity in quantity among the three adipose tissues were primarily glycerolipids and glycerophospholipids. Previous research has indicated that AF and SF are categorized as peripheral fats, with lipid molecules primarily dependent on hepatic synthesis and subsequently transported to the abdominal and subcutaneous regions through very-low-density lipoprotein (VLDL) [31]. Evidently, there are substantial differences in the lipid composition between IMF and AF or SF, whereas the distinctions between AF and SF are relatively minor. To explore the heterogeneity of lipid deposition in muscles compared to abdominal or subcutaneous tissue, an overlap analysis was conducted on differentially expressed lipid molecules. Among the upregulated differentially expressed lipids, 22 were found to be commonly present in both IMF vs. AF and IMF vs. SF comparisons, while among the downregulated differentially expressed lipids, 25 were shared between the IMF vs. AF and IMF vs. SF components. KEGG analysis revealed that upregulated DLMs in AF and SF are predominantly enriched in metabolic pathways and glycerolipid metabolism. On the other hand, in IMF, upregulated differential lipid molecules are mainly enriched in metabolic pathways and glycerophospholipid metabolism (Figure 3C,D).

### 3.4. Differential Genes of Different Adipose Tissues

To further elucidate y genes responsible for different lipid deposition, cDNA libraries for IMF, AF, and SF were constructed and sequenced. After rigorous filtering, AF and SF both yielded over 95% clean reads, and IMF achieved over 96% clean reads. FPKM was utilized as a measure of gene expression levels. Similar to the relationship observed in the lipidome samples, there are differences in transcriptional levels among different adipose tissues (Figure 4A). Comparative analyses were conducted between pairs of the three adipose tissues, utilizing DESeq2 for differential gene expression analysis with Benjamini–Hochberg correction. The criteria for DEGs were set at |log2Fold Change| ≥ 1 and *p* < 0.05. The comparison between AF and IMF resulted in 3369 DEGs, with 1743 upregulated and 1626 downregulated genes in IMF. Comparison between SF and IMF identified 5758 DEGs, including 1917 upregulated and 3841 downregulated genes in IMF. Comparative analysis between SF and AF revealed 131 DEGs, with 84 upregulated and 47 downregulated genes in SF. Figure 4B illustrates differences in gene expression between IMF, AF, and SF. The gene expression disparities between AF and SF appear to be relatively minor. Similarly, an overlap analysis was conducted on DEGs to explore the regulatory factors associated with the transcriptional differences in lipid molecules between muscle and other adipose tissue. Among the 1150 genes upregulated in IMF, the top five genes with the highest expression levels were ENSACDG00005013719.1, *PNPLA2*, *DCN*, *COL6A1*, and *FGL2*. Additionally, 1260 genes were simultaneously upregulated in AF and SF, with the top five genes being *FABP4*, *RBP7*, *APOA1*, *MRPS27*, and *GPX4*. Enrichment analysis of commonly expressed DEGs revealed that upregulated genes in IMF are primarily enriched in metabolic pathways and the MAPK signaling pathway. On the other hand, upregulated genes in AF and SF are mainly enriched in metabolic pathways and the PPAR signaling pathway (Figure 4C,D).

### 3.5. Identification of Potential Regulators of Different Adipose Tissues

Lipids, as a result of gene expression, can further elucidate the genetic basis of differential fat metabolism in various locations through the integrated analysis of lipidomics and transcriptomics. To comprehensively investigate the correlation between the combined lipidome and transcriptome datasets in IMF vs. AF and IMF vs. SF, a correlation analysis utilizing the quantitative values of genes and metabolites across all samples was conducted to identify potential regulatory factors. The pathway with the highest enrichment of commonly DEGs and DLMs was identified as metabolic pathways. Subsequently, correlation analysis between the DEGs and DLMs within this particular pathway was performed. Within the metabolic pathways, the lipid molecule PS (20:3_20:4) exhibited a highly positive correlation with the genes *PLCB1*, *DSE*, and *PDE5A*. Additionally, the lipid molecule TG (10:0_16:0_16:1) demonstrated a strong negative correlation with the gene *PLCB2* (|r| ≥ 0.8, *p* ≤ 0.05). Further analysis involved selecting the top 20 common DEGs with the highest expression levels and performing correlation analysis with the common DLMs. PG (19:2_20:4) showed a highly positive correlation with ENSACDG00005013719.1, while TG (10:0_16:0_16:1) exhibited a very strong positive correlation with *FABP4*. Conversely, TG (10:0_16:0_16:1) displayed a remarkably strong negative correlation with *LPL* (|r| ≥ 0.8, *p* ≤ 0.05). By selecting pairs with Pearson correlation coefficients greater than 0.60 and *p* < 0.05, an overlap analysis was performed, resulting in the identification of 23 pairs of DEGs and DLMs showing strong correlation. Among these, *NDUF4*, *GPX4*, and *DGAT2* genes demonstrated strong correlations with TG (10:0_16:0_16:1), PS (18:0_20:4), PS (20:3_20:4), TG (16:0_16:1_18:1), and other lipid molecules (Figure 5A–E).

### 3.6. Real-Time Fluorescence Quantitative Analysis

To corroborate the RNA-seq findings, a few genes associated with lipid metabolism were selected from each group as relatively important DEGs (AF vs. IMF: *MGLL*, *RBP4*, *PPP1R3B*, *ENPP1*, *SOX9*, *DUSP7*, and *THY1*. SF vs. IMF: *APOF*, *APOC3*, *CIDEC*, *ABHD11*, *FABP2*, and *LPL*. AF vs. SF: *FGF16*, *IGF2*, *IGFBP2*, and *WNT5B*). These genes were subjected to RT-qPCR analysis. The expression patterns of all these DEGs were consistent between the RNA-seq and RT-qPCR results, indicating that the RNA-Seq results were highly reliable (Figure 6A–C).

## 4. Discussion

IMF plays a crucial role in determining the quality of meat, while excessive deposition of AF and SF, particularly AF, during production can increase breeding costs. In this study, we identified heterogeneity in the quantity and content of lipid molecules through lipidomics in three adipose tissues. Subsequently, employing transcriptomics, we identified the key genes regulating fat deposition in different adipose tissues of geese. These results deepen our understanding of fat deposition in different adipose tissues of geese at both the metabolic and genetic levels, pinpointing potential regulatory factors for targeted fat deposition in specific adipose tissues, and contributing to molecular breeding to improve IMF deposition and reduce AF and SF deposition in geese.

Lipidomics, as a branch of metabolomics, utilizes liquid chromatography-tandem mass spectrometry (LC-MS/MS) for precise quantification of lipid molecules, enabling exploration of the inherent connections between lipid molecules and phenotypes [32]. In the present study, 839 lipid molecules were identified in the IMF, while AF and SF disclosed 827 and 837 lipid molecules, respectively, comprising 38 subclasses. TGs were identified as the predominant lipid class in adipose tissues. Through OPLS-DA analysis, lipid heterogeneity between different adipose tissues was identified. Compared to IMF, SF and AF exhibited higher abundance of GLs, with TG and DG as the major lipid subclasses. TG, an organic compound, is a glycerol ester synthesized by esterification of glycerol with three fatty acids. In poultry, TG synthesis primarily occurs in the liver through the glycerol 3-phosphate (Kennedy) pathway, initially forming DG and subsequently synthesizing TG under the catalysis of acyltransferases. TG serves as a crucial energy source for the body, releasing energy through the β-oxidation pathway. Excess energy intake, when greater than energy expenditure, results in the storage of surplus energy in the form of TG in the large lipid droplets of white adipocytes, making TG increase a major cause of fat deposition [33,34,35]. In contrast, IMF is predominantly composed of enriched GPs, with PC and PE as the major lipid subclasses. Phospholipids are essential components of cell membranes, possessing a hydrophilic head and hydrophobic tail, forming the basic scaffold of a phospholipid bilayer of cell membranes and participating in related signal transduction processes [36,37,38]. Phospholipids also serve as precursors for aromatic compounds contributing to the unique flavor of meat [9]. The generation of IMF adipocytes not only from mesenchymal stem cells (MSCs) but also from the differentiation of myoblasts, resulting in a higher number of low adipose maturation adipocytes in the IMF compared to SF and AF. Consequently, IMF exhibits higher phospholipid content [39,40]. This is consistent with previous reports indicating higher glycerophospholipid levels in the longest muscle of donkeys and upregulated DG in AF of chickens [26,41]. In summary, the GLs and GPs metabolic pathways are the primary pathways leading to fat deposition in different adipose tissue.

The heterogeneity in lipid deposition is a result of gene regulation. To elucidate the differential genes regulating fat deposition in different adipose tissues, we conducted transcriptomic analysis on the three fat adipose tissues [42]. Transcriptomic results revealed that 1150 genes were significantly upregulated in IMF, while 1260 genes were significantly upregulated in AF and SF. *FABP4*, ENSACDG00005013719.1, *LPL*, *PLCB1*, *DSE*, *PDE5A*, and *PLCB2* were selected as candidate genes strongly associated with fat deposition. *FABP4*, also known as adipocyte fatty acid binding protein, belongs to the lipid-binding superfamily of 14–15 kDa proteins [43]. *FABP4* can bind long-chain fatty acids and participate in lipid metabolism by transporting fatty acids [44]. It can transport fatty acids to mitochondria for β-oxidation and to adipocytes for triglyceride synthesis, thereby promoting fat deposition [45]. In this study, *FABP4* showed a strong positive correlation with TG (10:0_16:0_16:1) and was significantly upregulated in AF and SF. Previous studies have reported that adding folic acid to feed can reduce AF deposition, and the main reason for the reduction in AF percentage is the downregulation of fat synthesis-related genes such as *FABP4*, *C/EBPα*, and *PPARγ* by folic acid. In the longest muscle of pigs, the expression level of *FABP4* in the high IMF group was twice of that in the low IMF group, and *FABP4* showed a positive correlation with the number of fat cells and lipid content [46,47,48]. In contrast to *FABP4*, *LPL* showed a strong negative correlation with TG (10:0_16:0_16:1). Lipoprotein lipase (*LPL*), a member of the lipase gene family, is mainly present in muscle and adipose tissue, functioning as an acylglycerol hydrolase [49]. *LPL* primarily catalyzes the hydrolysis of triglycerides in chylomicrons (CM) and VLDLs, producing monoacylglycerols (MAG) and fatty acids for transport to other parts of the body for storage and utilization, thus influencing fatty acid β-oxidation and intramuscular fat deposition [50]. Depletion of *LPL* results in an increase in body weight and lipid accumulation in liver, skeletal muscle, and pancreas in mice [51]. Unlike the present study, the expression of *LPL* was positively correlated with IMF content in Baicheng broiler chicken [52]. This may be related to the regulation of triglyceride metabolism by *LPL*. In this study, *LPL* mostly utilized triglycerides for fatty acid oxidation to promote muscle growth and development.

Besides triglyceride deposition, the content of GPs also contributes significantly to IMF deposition. In this study, *PLCB1*, *DSE*, and *PDE5A* were strongly positively correlated with PS (20:3_20:4) and significantly upregulated in IMF. Studies have reported that these three genes are related to lipid metabolism. *PLCB1* (phospholipase) is correlated with marbling score and carcass traits in cattle, which might be an important candidate gene that increases deposition. *DSE* (dermatan sulfate epimerase) is a type of proteoglycan that can bind to low-density lipoproteins (LDLs) and be released by fat storage cells [53]. Although there is limited research in poultry and livestock, it has been demonstrated that the synthesis of a large amount of *DSE* in vascular smooth muscle cells can increase lipid accumulation [54]. *PDE5A* (phosphodiesterase) is not directly related to lipid metabolism, but it can participate in lipid metabolism by regulating cGMP hydrolysis [55]. In summary, *FABP4* may positively regulate abdominal fat deposition, *LPL* may negatively regulate IMF deposition, and *PLCB1*, *DSE*, and *PDE5A* may positively regulate IMF deposition. These may serve as potential targets for targeted IMF deposition.

To analyze the heterogeneity in fat deposition across different adipose tissue of Zhedong white geese, we conducted a correlation analysis of lipidomics and transcriptomics on goose IMF, AF, and SF. However, the experiment has certain limitations, as we selected geese of the same age at slaughter, neglecting the impact of body weight/carcass weight on fat deposition. Studies on sheep and pigs have indicated that live weight/carcass weight is a significant factor influencing fat deposition [56,57]. The geese used in our study had inconsistent carcass weights, introducing some variability. To mitigate the influence of carcass weight, further research with a more standardized approach is needed. Nonetheless, this study still holds significance in exploring the heterogeneity in fat deposition across different parts of geese.

## 5. Conclusions

In summary, the lipidomic profiles demonstrate that TG is the primary lipid subclass of different adipose tissues. However, GPs are the main lipid subclass deposited in IMF, while GLs are predominant in AF and SF deposits. DLMs are mainly enriched in glycerolipid metabolism and glycerophospholipid metabolism. Furthermore, transcriptomic analysis revealed differences in gene expression levels among different adipose tissues. Common DEGs are mainly enriched in metabolic pathways. Pearson correlation analysis identified regulatory genes involved in adipose tissue-specific deposition, among which FABP4 and LPL could serve as potential regulatory factors for molecular breeding targets. These findings provide new perspectives for understanding the heterogeneity of different adipose tissues and identifying targets for achieving targeted deposition of IMF, thus offering a valuable theoretical foundation for improving the quality of goose meat through breeding selection.

## Figures and Tables

**Figure 1 animals-14-01990-f001:**
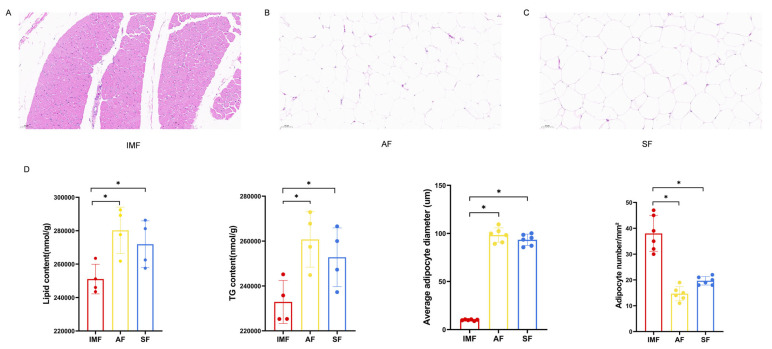
(**A**) Stained sections of intramuscular fat (IMF). (**B**) Stained sections of abdominal fat (AF). (**C**) Stained sections of subcutaneous fat (SF). (**D**) Lipid content, TG content, adipocyte diameter and adipocyte number were analyzed in three adipose tissues. Magnification of 20× was used. In the bar plot, * indicates *p* < 0.05.

**Figure 2 animals-14-01990-f002:**
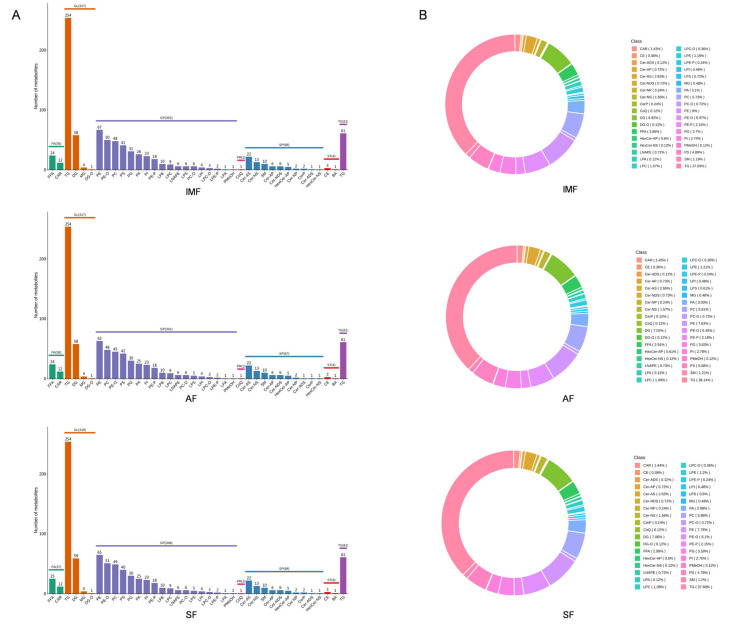
(**A**) Types and quantities of lipids in IMF, AF and SF of goose. (**B**) Comparison of lipid subclasses between different adipose tissues (nmol/g).

**Figure 3 animals-14-01990-f003:**
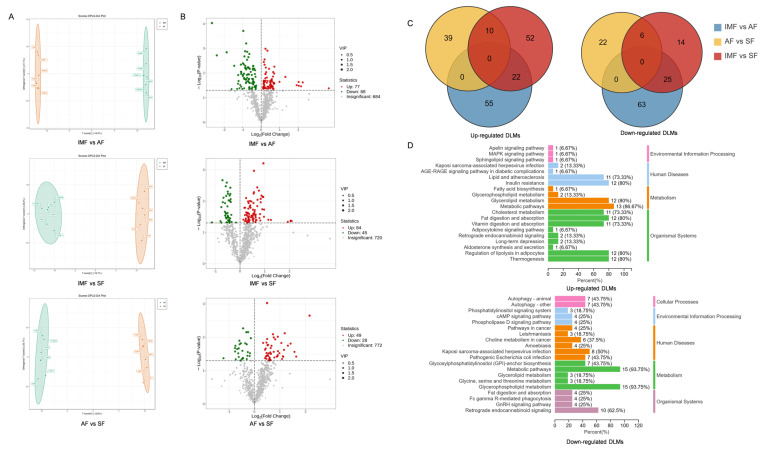
(**A**) Orthogonal partial least squares discriminant analysis (OPLS-DA) based on lipid molecules of IMF, AF and SF. (**B**) Volcano map of differential lipid molecules (DLMs) of IMF, AF and SF. (**C**) Venn diagram of the DLMs. (**D**) KEGG enrichment map of common DLMs in AF vs. IMF and SF vs. IMF.

**Figure 4 animals-14-01990-f004:**
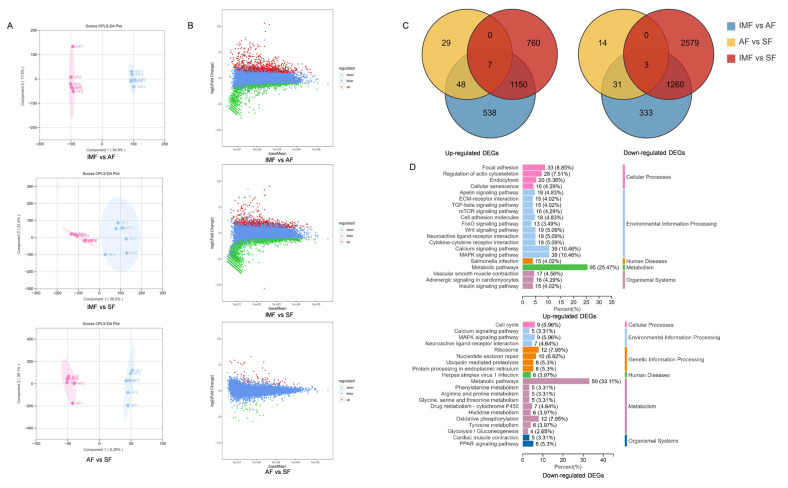
(**A**) Orthogonal partial least squares discriminant analysis (OPLS-DA) based on clean reads of IMF, AF and SF. (**B**) MA map of differentially expressed genes (DEGs) of IMF, AF and SF. (**C**) Venn diagram of the DEGs. (**D**) KEGG enrichment map of common DEGs between AF vs. IMF and SF vs. IMF.

**Figure 5 animals-14-01990-f005:**
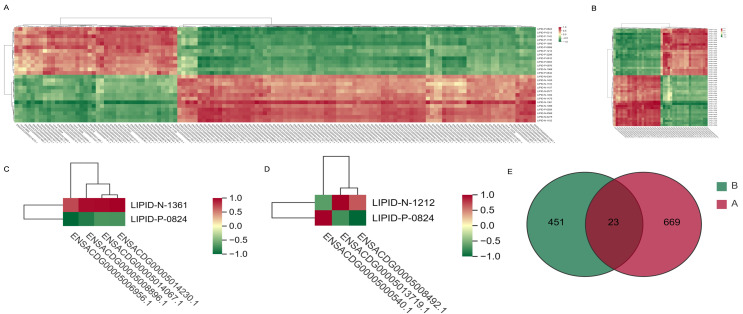
(**A**,**C**) Pearson correlation analysis between common DEGs in metabolic pathway and the main common DLMs in AF vs. IMF and SF vs. IMF (|r| ≥ 0.6, *p* ≤ 0.05 and |r| ≥ 0.8, *p* ≤ 0.05). (**B**,**D**) Pearson correlation analysis between the top 20 common DEGs and the main DLMs in AF vs. IMF and SF vs. IMF (|r| ≥ 0.6, *p* ≤ 0.05 and |r| ≥ 0.8, *p* ≤ 0.05). (**E**) Venn diagram of correlation analysis between the two groups (|r| ≥ 0.6, *p* ≤ 0.05).

**Figure 6 animals-14-01990-f006:**
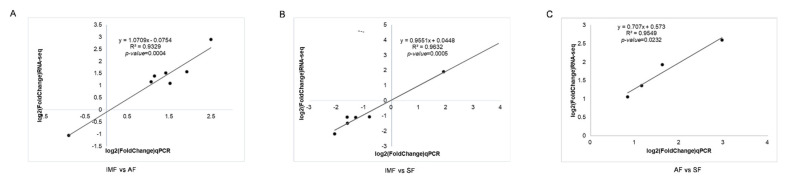
(**A**) Validation of differential gene in AF vs. IMF expression levels by RT-qPCR. (**B**) Validation of differential gene in SF vs. IMF expression levels by RT-qPCR. (**C**) Validation of differential gene in AF vs. SF expression levels by RT-qPCR.

## Data Availability

All data generated or analyzed during this study are included in this published paper.

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
