# Peer review of "Integrated Lipidomics and Transcriptomics Analyses Reveal Key Regulators of Fat Deposition in Different Adipose Tissues of Geese (Anser cygnoides)"

_animals, 2024, doi:10.3390/ani14131990_

Round 1

Reviewer 1 Report (Previous Reviewer 1)

Comments and Suggestions for Authors

The authors have significantly improved their manuscript in comparison to the first version submitted a couple of months ago. Redaction issues were corrected and now it is possible to read the article smoothly. Regarding my previous concerns regarding the thresholds that were used by the authors to deem a result as significant, this version contains references that help the authors to clarify why they selected such criteria and mitigate the concern of an arbitrary selection. Lastly, I consider that the new formulated conclusion is more appropriate, since now it is being supported by the results presented within the manuscript.

Within this version, I just have two observations that I would like to be solved:

1.- Figure 1A is not an histogram, is a bar plot, please correct the footnote.

2.- Within figure 6, could the authors please clarify why the P-values of the regression lines look as not significant? (e.g., P > 0.05). With such a high R2 values, I would thought that the P-values should be highly significant.

Comments on the Quality of English Language

The english redaction on this version of the manuscript has been significantly improved, now it is possible to read the manuscript smoothly.

Author Response

Reviewer 2 Report (Previous Reviewer 2)

Comments and Suggestions for Authors

Dear Authors,
Thank you for improving the manuscript; it reads better. Best regards,.

Author Response

Thank you very much for your kind work and consideration on publication of the manuscript “Integrated Lipidomics and Transcriptomics Analysis Reveal Key Regulators of Fat Deposition in Different Adipose Tissues of Geese (Anser Cygnoides)” (ID: animals-3084210) for publication. We appreciate your summary of the manuscript and encouraging comments. Meanwhile, we checked the manuscript again and revised it in detail. On behalf of my co-authors, we would like to express our great appreciation to you.

This manuscript is a resubmission of an earlier submission. The following is a list of the peer review reports and author responses from that submission.

Round 1

Reviewer 1 Report

Comments and Suggestions for Authors

Reviewer summary:

The manuscript’s objective was to elucidate the molecular regulatory mechanisms, heterogeneity, and specific deposition control targets of fat in different adipose tissues in goose meat in order to provide the scientific basis for efficient meat production on this species. Apparently, the authors wanted to generate information that ultimately can be helpful for the genetic improvement of these animals. The justification of the article was that fat deposition is positively correlated in different parts of the body, nonetheless, ideally what we would desire is that animals have a good capacity to increase intramuscular fat without accumulating too much abdominal and subcutaneous fat. Although the research idea is interesting and the authors employed multiple lab and analytic techniques to generate the results that are presenting, I consider that this manuscript has serious problems and flaws that preclude the acceptance of the article for publication. Please refer to my major and minor revisions below.

Major edits

I consider that the authors failed in thoroughly and clearly explaining the thresholds (or values) that they used to consider results as significant or important. For instance, even when the authors mentioned in the statistical analysis that the overall alpha value was 0.05, in Figure 1 (for example) they present results with 3 different significant values (0.05, 0.01 and 0.0001). Perhaps the authors idea was to highlight those results in which the P-value was smaller than 0.0001, but this is not needed and statistically redundant since as long as the P-value resulted smaller than 0.05 we would consider the result significant anyways. Another example is that authors explained that they used a Variable Importance in Projection (VIP) value of >1 to conduct the differential lipid molecule analysis, nonetheless, I did not identify a reference backing up this threshold, which make me thing it was set arbitrarily. In a similar way, for the identification of potential regulators of different adipose tissues the authors mentioned that they used a Pearson correlation coefficients greater than 0.60, nonetheless, there was no explanation of why this value was chosen which again, make me think that was established in an arbitrary way. Similarly, for the differential gene expression analyses, authors mentioned that the criteria for selecting differentially expressed genes (DEGs) were set at |log2Fold Change| ≥ 1 and P < 0.05, but no reference is backing this criteria up.

Another major flaw that I noticed on the manuscript is that authors struggled a lot to clearly write and express their ideas. The manuscript contains too many redaction issues that it gave me the impression that authors did not employ enough time to re-read the manuscript to improve it as much as possible before the submission to the journal. I consider this a major problem as it severely complicates the revision process since reviewers cannot completely focus on the scientific merit of the submission as we get distracted with the redaction errors that occur at almost every paragraph. Just to list some of the problems, there were multiple instances in where the authors used wrong words to describe a process (e.g., lines 136-137: the pectoralis major muscles, subcutaneous fat tissue, and abdominal fat tissue were “slaughtered”, here I believe the authors wanted to say that they took samples from this tissues..), also, many times the authors jumped from present to past tense in the same paragraph or they did not separated the words correctly. Additionally, many times the authors present an abbreviation that was not previously described, which of course makes difficult to immediately understand to what they are referring to. Moreover, many times the authors start a new sentence with a very long and complicated abbreviation crating a difficulty to read smoothly the article.

Some other problems that I identified were that some descriptions were missing in the materials and methods section. For instance, no description of the steps followed to conduct the histological analyses were explained, however, this is the first result that is presented. Also, within the statistical analysis, the authors mentioned that a specific software was used for “all the performance data”, nonetheless, it is not clear to which data they are referring to.

Lastly, even when the authors presented a plethora of results, I did not see where the authors explained how this information can be used to enhance the meat production of geese. In other words, I do not see a tangible application of the knowledge that the authors generated with all their analyses. It is not clear to me how the information presented in the manuscript can be used to conduct more sophisticated breeding practices in this specie to enhance intramuscular fat without increasing abdominal and subcutaneous fat.

It is because all of the previous elements that I consider that this manuscript cannot be considered for publication, it requires way more work from the authors side to enhance and correct redaction issues and also, to better explain the applicability of the knowledge generated.

Minor revisions

L2: I do not recommend using such type of abbreviations in the title of the manuscript as these are too technical and decreases the impact of the title itself... Rather, I suggest to focus of general and more understandable methods like lipidomic and transcriptomics

L14: Missing space between fat and parenthesis

L17: the word should be “affects” not “effects”. Also, the word “examines” is in present and should be in past

L18: explores à explored

L19: The aim was to elucidate valuable insights…

L26: Replace “in” by “between”

L27: lack of space (andSF)

L32: lack of space (elucidatedthe)

There are too many similar problems thorough the manuscript (lack of spaces, wrong words, problems with past and present tense)..

L76-78: I believe reference 18 is not needed as it pertains to chickens, not to goose. If there are reports in the species that you are working with, I suggest restricting the background information to that species.

Comments on the Quality of English Language

The manuscript contains too many redaction issues that it gave me the impression that authors did not employ enough time to re-read the manuscript to improve it as much as possible before the submission to the journal. I consider this a major problem as it severely complicates the revision process since reviewers cannot completely focus on the scientific merit of the submission as we get distracted with the redaction errors that occur at almost every paragraph. Just to list some of the problems, there were multiple instances in where the authors used wrong words to describe a process (e.g., lines 136-137: the pectoralis major muscles, subcutaneous fat tissue, and abdominal fat tissue were “slaughtered”, here I believe the authors wanted to say that they took samples from this tissues..), also, many times the authors jumped from present to past tense in the same paragraph or they did not separated the words correctly. Additionally, many times the authors present an abbreviation that was not previously described, which of course makes difficult to immediately understand to what they are referring to. Moreover, many times the authors start a new sentence with a very long and complicated abbreviation crating a difficulty to read smoothly the article.

Reviewer 2 Report

Comments and Suggestions for Authors

Dear Authors,

I enjoyed reading your MS 
titled: “Integrated UPLC-ESI-MS/MS-based Lipidomics and Transcriptomics Analyses Revealed the Key Regulators of Fat Deposition in Different Adipose Tissues of Goose (Anser cygnoides)”. The MS provides valuable information to the reader/other researchers and demonstrates the potentials of integrating advanced molecular techniques in uncovering the molecular mechanisms of fat deposition in avian species.  However, my major observation is that some references are missing or not cited in the text. For instance, reference “56” was not reported in throughout the MS but appears in the reference list. I strongly recommend the authors to thoroughly revise the references both in-text and on the reference list.

Great job.